# Present and Future of Bronchopulmonary Dysplasia

**DOI:** 10.3390/jcm9051539

**Published:** 2020-05-20

**Authors:** Luca Bonadies, Patrizia Zaramella, Andrea Porzionato, Giorgio Perilongo, Maurizio Muraca, Eugenio Baraldi

**Affiliations:** 1Neonatal Intensive Care Unit, Department of Women’s and Children’s Health, University of Padova, 35128 Padova, Italy; lucabonadies@hotmail.it (L.B.); patriz.zaramella@gmail.com (P.Z.); 2Human Anatomy Section, Department of Neurosciences, University of Padova, 35128 Padova, Italy; andrea.porzionato@unipd.it; 3Department of Women’s and Children’s Health, University of Padova, 35128 Padova, Italy; giorgio.perilongo@unipd.it; 4Institute of Pediatric Research “Città della Speranza”, Stem Cell and Regenerative Medicine Laboratory, Department of Women’s and Children’s Health, University of Padova, 35128 Padova, Italy; muraca@unipd.it

**Keywords:** bronchopulmonary dysplasia, extracellular vesicles, mesenchymal stem cells, mesenchymal stromal cells, neonatal ICU

## Abstract

Bronchopulmonary dysplasia (BPD) is the most common respiratory disorder among infants born extremely preterm. The pathogenesis of BPD involves multiple prenatal and postnatal mechanisms affecting the development of a very immature lung. Their combined effects alter the lung’s morphogenesis, disrupt capillary gas exchange in the alveoli, and lead to the pathological and clinical features of BPD. The disorder is ultimately the result of an aberrant repair response to antenatal and postnatal injuries to the developing lungs. Neonatology has made huge advances in dealing with conditions related to prematurity, but efforts to prevent and treat BPD have so far been only partially effective. Seeing that BPD appears to have a role in the early origin of chronic obstructive pulmonary disease, its prevention is pivotal also in long-term respiratory outcome of these patients. There is currently some evidence to support the use of antenatal glucocorticoids, surfactant therapy, protective noninvasive ventilation, targeted saturations, early caffeine treatment, vitamin A, and fluid restriction, but none of the existing strategies have had any significant impact in reducing the burden of BPD. New areas of research are raising novel therapeutic prospects, however. For instance, early topical (intratracheal or nebulized) steroids seem promising: they might help to limit BPD development without the side effects of systemic steroids. Evidence in favor of stem cell therapy has emerged from several preclinical trials, and from a couple of studies in humans. Mesenchymal stromal/stem cells (MSCs) have revealed a reparatory capability, preventing the progression of BPD in animal models. Administering MSC-conditioned media containing extracellular vesicles (EVs) have also demonstrated a preventive action, without the potential risks associated with unwanted engraftment or the adverse effects of administering cells. In this paper, we explore these emerging treatments and take a look at the revolutionary changes in BPD and neonatology on the horizon.

## 1. Background

Bronchopulmonary dysplasia (BPD) was first described by Northway [1] in 1967. Although more than 50 years have since gone by [2], this disease still lacks a comprehensive definition and, more importantly, a definitive cure.

Recent decades have seen enormous improvements in neonatal intensive care and in the development of innovative strategies that have led to a marked increase in the survival rates of preterm newborns, and especially those born at an extremely low gestational age. The burden of BPD has remained unchanged, however, even though the course of this disease has probably been improved. Despite prenatal and postnatal interventions, BPD remains the most common and severe chronic respiratory illness among preterm-born infants. Its incidence, defined as oxygen use at 36 weeks’ postmenstrual age or at discharge/transfer if before 36 weeks in infants who survived to 36 weeks, ranges from 15% to 35% among those born at < 32 weeks of gestation (WG), and has not changed over the past two decades. This result is the consequence of an improved survival rate; recently, many infants that would have died decades ago now survive developing BPD [3]. The challenges posed by BPD-related morbidity are consequently rising, as are the public health costs associated with prematurity and its comorbidities (neonatal sepsis, adverse neurodevelopmental outcomes, retinopathy of prematurity, and long-term respiratory complications extending into adulthood). The main cause of death in BPD patients is pulmonary hypertension, which is associated with four- to five-fold higher odds of infant mortality [4] and morbidity [5].

Taken together, BPD-related problems pose a huge and costly public health challenge, both before discharge and afterwards at home. Multiple services are involved in a BPD patient’s management (respiratory treatments, nutritional programs, neurodevelopmental stimulation, and so on [6]).

BPD is also a long-lasting disease, with effects that persist into adulthood [7]. A recent study examined the long-term effects of prematurity extending into adulthood (to age 30–45 years) in a Swedish national cohort of more than 4 million singleton live births. The authors reported an inverse relationship between mortality risk and gestational age. The higher mortality rate seemed to be mediated by conditions such as diabetes, and cardiovascular and respiratory diseases. They concluded that preterm birth should be recognized as a chronic condition that requires long-term follow-up for the prevention and treatment of potential health sequelae into mid-adulthood [8]. BPD per se is related with higher mortality and morbidity, highlighting how former preterm BPD patients should be followed with even more care.

Hence, there is a strong interest in the feasibility of developing new prevention or treatment strategies for children born preterm.

This review outlines our present understanding of the future prospects for the prevention and treatment of BPD, based on our knowledge of the disease’s pathogenesis, and on the current management of extremely low birth weight (ELBW) babies. This is not intended to be a systematic review of the literature. Articles were identified by searching PubMed using Medical Subject Headings, without using any selection or rejection criteria. Review articles are also cited.

## 2. Definition and Diagnosis

Bronchopulmonary Dysplasia is a chronic respiratory disease that affects a significant fraction of former extremely premature infants, this disease is a heterogeneous condition that develops on an extremely preterm lung exposed to different pathogenetic noxae.

The most often-mentioned consensus on how to diagnose BPD comes from the NICHD 2001 [9]. This definition considers two criteria: weeks of gestation (WG); and the need for supplemental oxygen beyond 28 days of life and/or oxygen dependence at 36 weeks post-menstrual age (PMA). To improve on this diagnosis, a physiological measure, like the “room challenge test”, was suggested [10].

In 2018, the NICHD proposed a revision [11] of their definition, considering newer methods of noninvasive ventilation, a reclassification based on grades, and radiographic evidence of pulmonary parenchymal disease.

Though useful to neonatologists, these definitions do not seem to predict long-term outcomes. Those caring for patients with BPD are more interested in establishing new diagnostic criteria that can shed light on a patient’s likely outcome. They also seek a more comprehensive definition of the disease, based on accurate clinical pathophysiology and the identification of biomarkers, rather than on interventions [12]. An international neonatal consortium [13] recently emphasized the concept that a definition of BPD should be clinically meaningful and strongly associated with the subsequent development of respiratory problems. The long-term effects of BPD are not all reached by 36 weeks after conception, making this age a less important milestone for the purposes of BPD stratification.

For this purpose, Isayama et al. [14] explored different timepoints to better diagnose BPD and predict respiratory outcomes, identifying the need for oxygen at 40 weeks PMA as the best predictor for serious respiratory morbidity, leaving an open field to the debate not only on which clinical data but also on which timepoint to be used.

Jensen et al. [15] recently reviewed the 18 existing definitions of BPD with a view to identify the most appropriate one for predicting childhood morbidity through 18–26 months. They found that the best definition (correctly predicting death or severe respiratory morbidity in 81% of cases) classified disease severity on ventilatory support at 36 weeks PMA alone, regardless of any supplemental oxygen use.

As Thebaud et al. [16] explained, a definition’s predictive value can be improved by means of better objective measurements and biomarkers of lung injury and BPD, and by incorporating antenatal factors, such as intrauterine growth restriction (IUGR), a mother’s hypertension or smoking, and male sex of her offspring. A single definition is unlikely to be able to address all the needs of neonatologists and pediatricians, caregivers, pharmacists, parents, and industries, nor is it likely to point towards significant endpoints. However, a more appropriate general definition would provide a better description of patients with BPD and enable more targeted preventive therapies.

## 3. Pathology

In infants with BPD, the pathological findings [17] include fewer secondary septa and alveoli, and sites of emphysema, meaning a smaller alveolar surface area, which is crucial to proper gas exchange. Angiogenesis is impaired with dysmorphic vessels and capillaries. Thickening of the muscle layer of the arterioles prompts an increase in vascular resistance and pulmonary hypertension. More fibrotic tissue is formed, with a widening and thickening of the interstitial spaces (Figure 1). This picture differs slightly from earlier descriptions (before surfactants became available), such as Northway’s [1] report of injury, inflammation and fibrosis caused by volutrauma or barotrauma, and oxygen toxicity. Today’s “new” BPD is the result of less severe injury in very immature lungs [18], but very little is known about the pathology of this “new BPD”—partly because fewer autopsies are being performed.

Angiogenesis and vessel branching reportedly drive alveolar growth in the lung, so the dysmorphic vascular architecture and angiogenesis described in BPD patients are important. Vascular channels have been described in infants who died of BPD. These vessels run through lobular sites normally occupied by veins towards the pulmonary arteries, then to the vasa vasorum of the pulmonary plexus and bronchial venous plexus [19]. These changes presumably provide the anatomical grounds for pulmonary hypertension in BPD, but these shunts may also serve as a “pop-up” valve to reduce the severity of pulmonary hypertension.

More collaboration between neonatologists and pathologists, a greater awareness of the importance of conducting autopsies on deceased BPD patients, high-resolution imaging, and increasingly reliable animal models could help us to shed more light on the still dark side of BPD.

## 4. Pathogenesis

A better understanding of how the premature infant’s lung develops and of the pathogenesis of BPD could drive the search for novel treatments. Lung growth appears to be an intricate, highly orchestrated process involving multiple cell lines, and is guided by numerous different signaling pathways, which could all be disrupted by factors implicated in the onset of BPD [20].

At birth, most preterm newborns are in the saccular and canalicular phases of lung morphogenesis (assuming a viability limit at 24 weeks PMA). There have occasionally been reports of survivors born at 22 WG coping with breathing even during the canalicular phase of lung development [21].

Given the cellular architecture of an underdeveloped lung, low gestational age is the most significant determinant correlating with the onset of BPD. The premature lung has several features that predispose it to BPD, in addition to the well-known lack of surfactant. It also has: less-developed skeletal airway structures (extracellular matrix, collagen or elastin); less-developed antioxidant mechanisms; a lower compliance; and inadequate fluid clearance [22]. All these features make the lung more vulnerable, but many other factors contribute to causing BPD in such vulnerable tissue. BPD has a multifactorial etiology, with prenatal and postnatal mechanisms causing inflammation and injury. The consequent disruption of the lung’s development also involves an aberrant repair mechanism.

## 5. Prenatal Risk Factors

One significant variable contributing to the risk of BPD is intrauterine growth restriction (IUGR) [23,24]. The mechanisms behind this phenomenon seem to include an impaired growth of terminal airways and gas exchange units, but also a weaker expression of surfactant protein and mRNA, and possibly also the induction of an increased inflammatory response [24], but the mechanism underlying the correlation between BPD and IUGR is quite complex and still not completely understood. Bose et al. [25] suggested a multiple pathogenetic mechanism involving reduced lung growth, impaired angiogenesis, chronic fetal hypoxia with TGF-beta upregulation, and surfactant mRNA reduced expression. The hypothesis recently gaining importance is the “vascular hypothesis”, as initially suggested by Abman [26] and recently widely explored [27].

Preeclampsia also seems to be related to a greater risk of BPD [28], although the precise mechanism of disruption involved is not known and could be indirectly consequent to fetal growth restriction due to preeclampsia and not directly to the latter. One hypothesis relates to an imbalance in angiogenic factors, and consequent impaired angiogenesis, potentially having a significant role, contributing to the abnormal structure and distribution of the distal microvasculature in the lungs of infants with BPD [29,30].

Looking at preeclampsia, the EPIPAGE−2 Cohort Study [31] showed how placental-mediated pregnancy complications with fetal consequences are associated with moderate to severe BPD in very preterm infants, but isolated maternal hypertensive disorders are not. This suggests again how fetal growth restriction could be the principal mechanism involved in BPD development.

Some authors have suggested that chorioamnionitis is associated with BPD, as it would give rise to higher concentrations of proinflammatory cytokines in the amniotic fluid [32]. This would mean that the pathogenesis of BPD begins already in utero, in the choriodecidua, and untargeted metabolomic profiling of the amniotic fluid would seem to confirm such a hypothesis [33,34]. The role of chorioamnionitis in the onset of BPD is still debated, however [35].

Morrow et al. [36] evidenced how maternal smoking during pregnancy increased the odds of having an infant with moderate or severe BPD by 2.02-fold, suggesting increased inflammatory cytokine production, altered placental function, or direct impact on lung development, which disrupts lung structure and function as possible involved mechanisms [37,38]. Maternal smoking results also related to a higher risk of late respiratory disease in former preterm babies [36], defined as any of the following events occurred over the first 2 years of life: one or more respiratory hospitalizations; use of inhaled steroids, inhaled bronchodilators, and/or diuretics; and a physician’s diagnosis of asthma, reactive airway disease, or a BPD exacerbation.

Multiple pregnancy has been excluded as a risk factor for BPD [39,40], but a high concordance between twins has been described [41].

Among the extremely low birth weight (ELBW) newborns, some cases may be similar and have similar histories, but not all of them will develop BPD. This would suggest a genetic predisposition, a possibility supported by evidence of monochorial twins having a significantly higher concordance in the incidence of BPD than bichorial twins: up to 79% of this variance has been attributed to genetics [41,42]. Such genetic factors have been investigated mainly in gene ablation studies, in relation to lung branching morphogenesis [43,44], in models of lung disease [18], and—more recently—in genome-wide association studies and exome sequencing on preterm-born infants. Multiple genes have been examined and proposed as BPD risk factors, but no clear genetic target has been identified so far [45,46]. A recent wide exome sequencing (WES) study [47] has been able to identify some gene variants that could be suitable to be evaluated as BPD predisposing factors, but more importantly, all the associated pathways could become future targets for therapy and prevention of BPD.

## 6. Demographic Risk Factors at Birth

Extreme prematurity and extremely low birth weight, associated with a small and underdeveloped lung, are obvious risk factors for BPD.

In addition, the mode of delivery seems to have been identified as related with a different incidence of BPD, with C-sections seeming to be related to a reduced rate of BPD [36]; a possible explication is the influence of vaginal flora on the microbioma [48], but this relationship is not always confirmed [49] and it was not protective against late respiratory disease [36].

Male sex also seems to correlate with a higher risk of BPD than in females [50,51]. Some animal studies have explained this with a higher level of inflammation, and a more severe loss of alveolarization and angiogenesis in males [52].

Ethnicity seems to influence the risk of developing BPD too, with black race carrying a lower risk of BPD but a higher risk of persistent respiratory morbidity [53].

## 7. Postnatal Risk Factors

A particularly fascinating field of investigation concerns how lung tissue healing processes may mimic intrinsic ontogenic phases to reproduce the physiological machinery. Having only immature or no antioxidant defenses [54] exposes the preterm newborn to BPD because they are likely to need oxygen supplementation; this prompts an excessive production of cytotoxic reactive oxygen metabolites, which overcome the antioxidant system [55].

Hyperoxia is the best-known and most often-studied postnatal risk factor for BPD. Its pathogenesis includes epithelial and endothelial cell death, inflammation, impaired mitochondrial function [56], and reduced numbers of resident lung and systemic progenitor cells [57,58].

Mechanical ventilation and volutrauma or endotrauma have been identified among the main causes of BPD, as a result of bronchiolar hyperinflation and injury [59] leading to the activation of inflammatory signaling in resident lung cells [60], inducing the recruitment of inflammatory cells [61], and altering the pathways involved in alveolarization [62]. This association has prompted neonatologists to change their ventilation strategies, starting already in the delivery room. Lower tidal volumes and less invasive ventilation techniques are now used to help protect the infant’s lung.

Another suggested pathway involves late surfactant deficiencies, but recent studies testing the late administration of surfactant have shown little or no benefit [63]. This might be because multiple mechanisms are implicated, such as a decreased de novo surfactant production, an increased degradation of intra-alveolar large aggregate surfactant, a loss of alveolar surfactant secondary to phagocytosis, and a diminished recycling of surfactant phospholipids [64].

Sepsis is another factor associated with a higher risk of BPD, especially in infants with candidemia [65], an infection that naturally activates the inflammatory cascade with a consequent production of pro-inflammatory cytokines, the migration of PMNs, and changes in vascular permeability, all of which promptly cause alveolar damage, and long-term alterations as well. It is important to emphasize that a common, and often underestimated infection like *Staphylococcus epidermidis* also induces pro-inflammatory responses in human alveolar epithelial cells [66].

This risk of BPD is amplified by a symptomatic patent ductus arteriosus (PDA) [67], even though the pathogenic mechanism behind this association is still unclear.

The microbiota is an important emerging field of interest; the fetal lung has probably its own microbiota since fetal life, seen that placenta and amniotic fluid harbor their own microbiota [68]. Lung microbiota, influencing the homeostatic control and development of immune system [69], could influence also the inflammatory and tissue repair responses to pathogenetic insults. Recently, an alteration of lung microbiota, specifically a reduction in its diversity at birth, seems related with BPD development [70,71]. Another possible mechanism through which microbioma can influence lung development is microbial metabolite production altering inflammatory response [72,73].

Extrauterine growth restriction (EUGR) correlates with a higher risk of BPD too [74,75], and this association is theoretically attributable to a reduced lung growth as a result of a reduction in body growth and its mediators. Postnatal growth restriction is quite common in preterm infants; its presence and its combination with hyperoxia result in decreased expression of key modulators of angiogenesis and vascular tone including VEGF, VEGF receptor 2, HIF1α, HIF2α, eNOS and NOS metabolites [74], again suggesting the importance of the vascular hypothesis in BPD development, but many other mechanisms could be involved [76]. To ensure a good nutritional intake will so result in a reduced risk of BPD, but also improve neurodevelopment [77].

All these risk factors take effect by altering various pathways that contribute in some way to the development of BPD.

## 8. Cellular Modifications

One of the intriguing aspects of BPD concerns the role of inflammatory cells in its pathobiology. Macrophage polarization causes a paradoxical derecruitment of M1 vis-à-vis M2 cells. M1 macrophages have proinflammatory functions, while M2 macrophages are involved in healing and repair pathways. Macrophage phenotype seems to be associated with disease severity in preterm infants with BPD [78,79]. Both the choriodecidua and lung response appear to have a part to play as well, judging from findings in tracheal aspirates from preterm newborns [80]. Neutrophils act as host defense cells, reacting to injury caused by oxidative species, cytokines or elastases [81]. They have also been studied more recently for their role in tissue repair through defensins and elastases: such an action in the lung of mucosa-protecting effect of neutrophils is now being investigated [82].

Another area of investigation is the system of pulmonary neuroendocrine cells (PNECs): changes in the number of these cells, and in their peptide levels, have been reported in infants with BPD [83,84], and in rats exposed to hyperoxia [85]. The role of PNECs, and of drugs acting on their serotonin production, deserves further study. Higher levels of bombesin-like peptide, and of calcitonin- and serotonin-immunoreactive PNECs have been reported in infants dying of BPD [83].

One working hypothesis is that exhaustion or dysfunction of the lung’s resident stem/progenitor cells contributes to lung growth impairment in BPD, and to the immature lung’s inability to repair itself [86,87,88].

## 9. Growth Factor Alterations

All the above-mentioned (cellular and mediator) mechanisms have to do with the inflammasome, which is crucial to the development of BPD. There have been several reports of proinflammatory and chemotactic factors being detected in higher or lower concentrations in infants developing BPD [89,90]. In particular, certain cytokines (IL−6, IL−8 and GM-CSF) seem to be directly involved in early lung injury evolving into BPD. Transforming growth factor β (TGF-β) is implicated in elastogenesis and alterations in this process due to an inflammatory stimulus have an antiproliferative effect on alveolarization [91,92]. Leroy et al. [93] demonstrated that systemic inflammation occurs early in the neonatal period in infants with BPD, before any clinical symptoms become apparent.

Growth factors are heavily implicated in the pathogenesis of BPD. In particular, lower serum levels of insulin-like growth factor−1 (IGF−1) relate to the onset of BPD [94,95,96,97], however high the tissue levels of IGF−1 and its receptor IGF−1R might be in the peribronchial and perialveolar mesenchyme [95,98,99]. IGF-1 could be said to behave like a hormone in the general circulation, but like a cytokine at local level. Systemic IGF−1 is a growth stimulant and growth marker. One hypothesis for its link with BPD is that IGF−1 is a vascular endothelial growth factor (VEGF) stimulator [100], with a possible involvement in lung angiogenesis. Another possible mechanism is that low levels of this hormone are related to extrauterine growth restriction (EUGR), which is a risk factor for BPD. On the other hand, IGF−1 is locally involved in the lung injury/repair process, enhancing the local proliferation of human fetal lung fibroblasts, and stimulating collagen production—both of which are part of the aberrant lung repair process.

Past and present studies are revealing the crucial role of lung angiogenesis: as it actively promotes alveolar growth, an altered angiogenesis (i.e., high levels of endostatin or bombesin-like peptides) probably plays a part in the pathogenesis of BPD as well [101,102,103]. Researchers have identified a suppressed vascular growth in the early stages of BPD, and an excessive, dysmorphic (probably compensatory) growth in later stages, which would pave the way to pulmonary hypertension [104,105]. VEGF is one of the main elements involved in angiogenesis, and consequently, in lung development. Its inhibition seems to impair angiogenesis and distal airspace growth, and to cause pulmonary hypertension in neonatal and infant rats [106]. The levels of this growth factor are distinctly elevated in airway aspirates obtained in the first days of postnatal life from preterm infants who later develop BPD [107], but blood levels of VEGF seem to be lower than normal in infants developing BPD [108].

All the above-described potential mechanisms go to show that BPD is not really a single disorder, but a set of disorders (all of which involve a disrupted alveolar and vascular growth), and different factors may contribute to its development in different patients at risk. All these factors contributing to a perturbed lung development may also follow different paths and carry a different weight in the progression of BPD in different individuals.

## 10. Animal Models and Morphometric Methods

Various experimental animal models of BPD have been developed in an effort to improve our understanding of its pathogenesis and assess possible therapies. These models have included: postnatal exposure to hyperoxia of term-born mice [109], and rats [110], and preterm or term-born rabbits [111]; and chronic artificial ventilation of preterm lambs [112], and preterm baboons [113]. As stated by Nardiello et al. [114], these efforts have pointed to an increasing need for a standardized model. In experimental studies addressing strategies for the prevention and treatment of BPD, it is pivotal to use reliable and unbiased morphometric methods to quantify alveolarization. Radial alveolar counts (RAC) and mean linear intercepts (Lm) are frequently adopted as surrogates for alveolar size and alveolar number, but these methods are not completely unbiased because they are calculated in surfaces and not in three-dimensional probes. That is why the latest standards for quantitatively assessing lung structure (such as those of the American Thoracic Society/European Respiratory Society) stress the importance of using unbiased stereological methods [115]. The main stereological parameters for assessing alveolarization are: total volume of alveolar air spaces; total number of alveoli; mean alveolar volume; total volume and surface area of alveolar septa; and mean alveolar septal thickness [116,117,118]. Recently, automatically derived parameters have been evaluated in comparison with stereological analysis, demonstrating a comparable efficiency in detecting changes in alveolarization [119], with a fractal approach that may be fit to this automatic analysis [118].

## 11. Prevention and Management

Ultimately, the most effective strategy for preventing BPD is to avoid extreme preterm birth [2]. Where this is impossible, the current approach to BPD includes numerous prevention strategies, albeit of limited efficacy in this respiratory disease. Some routinely used therapies are listed below (Table 1).

Together, these measures have reduced the BPD-related mortality rates among preterm infants, but not the incidence of BPD [3] (because of the rising numbers of infants at risk of this disease). Hence, the ongoing need to seek innovative approaches and more effective therapies for BPD.

## 12. Long-term Pulmonary Outcome

Physiologically, lung function tends to increase gradually throughout childhood and adolescence, reaching a plateau at around 23–25 years of age [146]. It is now clear that people entering adulthood with a lower than normal lung function are at a higher risk of developing chronic obstructive pulmonary disease (COPD) in later life. That is why it is so important to contain early neonatal lung injury [22,147]. BPD and prematurity are clear examples of how an insult occurring in the perinatal period can be associated with health issues that persist into adult age [147,148].

Even if we lack the ideal technology for assessing lung function in infants, several cross-sectional studies have studied BPD survivors of school age and older, documenting their airway obstruction [149,150,151]. An impaired lung function has been identified in both “old” and “new” BPD survivors, and it is associated with more respiratory symptoms and abnormal chest imaging findings [152]. In addition, a deterioration of lung function during childhood and adolescence has been described in longitudinal studies of preterm and BPD survivors [153,154,155].

Our group recently published the results of the “Padova BPD study” [156] begun in 1991 (at the start of the “surfactant era”) to monitor lung function in BPD survivors longitudinally from birth to early adulthood. This study showed that significant airway obstruction in infancy persisted at 24 years of age, associated with a failure to achieve an optimal peak lung function in the adult. Being based on data from birth onwards, these findings support the concept of BPD as the earliest-onset and longest-lasting obstructive pulmonary disease in humans, which could predispose to COPD in adulthood [7,156].

COPD is commonly considered a smoking-related condition in so-called “susceptible” smokers, but about one in three COPD patients have never smoked. Nowadays, COPD is seen as a syndrome with different phenotypes and natural histories. Factors taking effect at different times early in life, such as abnormal lung growth, maternal smoking, childhood asthma, and early respiratory infections, may impair lung function and raise the risk of COPD, and strategies to avoid them can help prevent its development in adult life [157].

A relevant question is whether the long-term pulmonary consequences of prematurity and BPD depend essentially on stable structural damage, or also reflect an ongoing, active airway disease [158]. For the time being, there are no pathological data available to shed light on the characteristics of BPD survivors’ lung tissue beyond infancy. In a recent case series of histopathological findings in bronchial biopsies from adolescent survivors of BPD, the authors reported finding lymphocytic infiltrates and signs of immature neo-angiogenesis, pointing to an ongoing active inflammatory process in their airways [159].

## 13. BPD Treatment: Prospects and Needs

### 13.1. BPD Biomarkers

All promising treatments have to be tested, then used only for patients likely to benefit from them. This means that we need to identify patients likely to develop BPD as early as possible. Then, we need to establish which pathogenic mechanisms are the main culprits in the single infant in order to provide the most appropriate treatment in each case.

A reliable biomarker would have to be detectable in the early phase of the disease, and thus, enable prompt intervention to prevent or minimize its detrimental effects. The cellular and blood/fluid markers tested to date do not fit the bill because they have shown a low predictive value, adding minimal power to clinical variables alone [160].

The so-called “-omic” techniques have clearly brought advances in the detection of early biomarkers of BPD, and several promising studies have been published [161,162]. These technologies have yet to pinpoint a gold-standard biomarker for use in the routine clinical care of infants at risk of BPD however [163]. Using proteomic screening in preterm infants, one study recently found a validated set of proteins (SIGLEC−14, BCAM and ANGPTL3) capable of identifying BPD soon after birth and pointing to key processes in the disease’s subsequent development [164]. Another study applied metabolomics to the analysis of amniotic fluid, and the authors suggested that this approach might be useful for predicting BPD [33]. Such -omic techniques have the potential to uncover new developmental pathways that may help us to find new targets for therapy.

### 13.2. Other BPD Predictors

A web-based BPD estimator was recently developed by NICHD to identify infants at risk of BPD who may benefit from postnatal steroids [50,165].

Though it is not a biomarker, lung MRI has recently revealed a good capacity for predicting short-term outcomes in the setting of BPD [166]. Lung ultrasound has also proved capable of predicting the onset of BPD [167].

In future, these tools (like biomarkers) can probably help pinpoint patients at higher risk of BPD, and also, enable us to improve on its definition and the scoring of its severity.

## 14. New Horizons

### 14.1. Corticosteroids

We have known for years that corticosteroids are effective in improving respiratory function in preterm infants, but these drugs have recently been put to a new use [168]. Their systemic use has been considerably reduced in the last decade, owing to fears over side effects, and especially possible neurological consequences, such as a higher risk of cerebral palsy [169]. Recent evidence has tended to debunk such worries and encourage a cautious, more targeted use of systemic steroids, though a large-scale RCT will be needed to clarify the best posology [170]. Another reason for a renewed enthusiasm for steroids lies in results obtained recently with their prophylactic use [171,172].

Another approach that has been explored involves the intratracheal administration of corticosteroids (in an effort to avoid their unwelcome effects). A recent paper reported encouraging results of administering intratracheal budesonide mixed with surfactant [173]. At a time when we may still have our doubts about the use of systemic corticosteroids, this targeted therapy could prove a promising, readily accepted and effective strategy [174]. McEvoy et al. [175] evaluated the optimal safe dose of this treatment, showing a lung-targeted anti-inflammatory effect with a tenth of the previously used dose, with minimal systemic metabolic effects, and evidenced a systemic passage of the treatment, slightly reducing the enthusiasm about targeted therapy.

With the same goal in mind, aerosolized steroids have been the object of numerous studies. In one recent report, early use (within 24 h) of inhaled budesonide in extremely preterm newborns was found associated with a lower incidence of BPD. At two-year follow-up, the study found no differences in terms of neurodevelopmental disability, but the mortality was higher in a steroid-treated group [176,177,178]. However, the dose of inhaled corticosteroids used in this study was very high.

### 14.2. Growth Factors

Looking at the pathogenesis of BPD, growth factors (or their agonists and antagonists) would seem to have therapeutic potential. When the use of IGF−1 supplementation to prevent retinopathy of prematurity was examined, it achieved the encouraging secondary outcome of a reduction in the incidence and severity of BPD [96]. This led to the suggestion of a possible new therapeutic approach that is currently the object of a study by Takeda-Shire [179].

As for the involvement of angiogenesis in the development of BPD, the antenatal or postnatal inhibition of VEGF with a monoclonal antibody against soluble fms-like tyrosine kinase (sFlt−1) prompted an improvement in lung structure in an animal model of BPD [180].

The already cited wide exome sequencing study [47] could offer some new perspectives in this field, unraveling future new pathways to evaluate and target.

### 14.3. Mesenchymal Stem/Stromal Cells and Extracellular Vesicles

Several studies on BPD found a reduction in the population of resident stem cells in patients’ lungs [86,87,88]. This prompted the suggestion that such a phenomenon could have a role in the pathogenesis of BPD, and that replacing this cell population might help reverse the course of the disease. Mesenchymal stem/stromal cells (MSCs) have been investigated as a potential tool for preventing and treating many lung diseases. MSCs were originally described as stem cells, but it was later recognized that data to support such a functional designation are still lacking, so the term “stromal” was recommended by the International Society for Cellular Therapy [181]. MSCs are plastic-adherent cells with a multipotent differentiation capacity in vitro. They can be isolated from a variety of tissues, including bone marrow, fat, cord blood and tissue, and placenta [182]. They can modulate immune response, promote angiogenesis, and enhance regeneration and repair to protect tissues against a variety of injuries. These properties make MSCs an attractive therapeutic tool in regenerative medicine, particularly in pediatric diseases [183,184].

Kourembanas’ group [185] was the first to demonstrate the beneficial effects of intravenous MSCs or MSC-conditioned medium (CM) in an animal model of hyperoxia-induced BPD. Their promising results were confirmed by other investigators using intravenous, intratracheal [186,187,188,189,190,191,192,193,194] and, more recently, intraperitoneal [195] and intranasal [196] modes of administration. These treatments lowered the levels of inflammatory mediators like IL−6 and TNF-α [197], and the expression of angiotensin II, angiotensin II type 1 receptor, and angiotensin-converting enzyme [198], which are involved in the development of BPD. MSCs also improved alveolar structure and angiogenesis, attenuating lung fibrosis and inflammation, and increasing exercise capacity in animal models of BPD [186,187,190].

Intriguingly, the intratracheal administration of MSCs derived from human umbilical cord blood was found to attenuate both lung and brain injuries in rat pups exposed to hyperoxia [199], suggesting that local administration can have systemic effects. O’Reilly et al. [200] recently described how multiple doses of MSCs improved BPD-like lung injury, both early on and at a later stage of the disease.

Only two translational trials have been published to date on the use of MSCs for treating BPD in humans [201,202]. In a phase I dose-escalation study, Chang et al. [201] tested the safety and feasibility of intratracheal MSCs in nine preterm infants at high risk of BPD. The treatment was well tolerated and without any serious adverse effects. As a secondary endpoint, this study also reported some preliminary evidence of the treatment’s efficacy, since the IL−6, IL−8, MMP−9, TNF-α and TGF-β1 levels in tracheal aspirates were lower after the treatment, and MSC recipients seemed to have less severe BPD. It should be noted, however, that the time course of inflammatory cytokine production in BPD is not known, and the clinical comparison was drawn with a historical case-matched group. A two-year follow-up study confirmed the lack of any long-term side effects in the babies treated with MSCs [203]. Powell et al. [202] performed a similar phase I dose-escalation trial on the safety and feasibility of intratracheal MSCs in twelve preterm infants at high risk of BPD. Here again, the treatment was well tolerated and appeared to be safe and feasible. This study did not include a control group, however, and the authors voiced the need for a larger, blinded RCT.

Similar beneficial effects were seen after administering other types of cell, including cells derived from human amniotic fluid [204,205,206], and epithelial progenitor cells. The safety, tolerance and feasibility of the intravenous administration of human amnion epithelial cells (hAECs) were recently tested by Lim et al. [207] on six infants with BPD, finding no adverse events in the short-term, or at two years of age [208].

These pioneering studies have generated great interest in cell therapy for the prevention of BPD. Table 2 provides a list of similar recent and ongoing studies.

MSCs were initially believed to differentiate and repopulate injured sites with tissue-specific cell phenotypes, but it soon became evident that the therapeutic effects observed were mediated by paracrine signals regulating immune response, counteracting apoptosis, limiting fibrogenesis, stimulating endogenous stem cells, and thus, resulting in tissue repair [209,210]. In the last decade, accumulating evidence has shown that such signals are conveyed mainly by various membrane vesicles, including exosomes and microvesicles, which are collectively known as “extracellular vesicles” (EVs) [211,212,213]. EVs are complex biological machines secreted by all cell types. They range from 0.03 to 1 micron in size. They are distinguished mainly by their biogenesis, since they exhibit overlapping physical and chemical characteristics. EVs can affect both the metabolism and the phenotype of target cells by delivering their cargo of proteins, miRNA, mRNA, DNA, and even mitochondria. The immune modulating and pro-regenerative effects of MSC-derived EVs have been confirmed both in vitro [214,215,216] and in several animal models of disease [217,218,219], but the mechanism of action of EVs remains largely a mystery. Part of their therapeutic effect seems to be mediated by tumor necrosis factor alpha-stimulated gene−6 (TSG−6) [220] and VEGF [221]. On the other hand, as Willis et al. [222] noted, EV treatment results in pleiotropic effects on genes associated with hyperoxia-induced inflammation and immune responses. One such effect is to modulate the macrophage phenotype fulcrum, suppressing the proinflammatory “M1-like” activation and switching to an anti-inflammatory “M2-like” state. This futuristic therapy has aroused great interest because of the growing evidence of EVs isolated from MSC-conditioned medium having a therapeutic effect in experimental models of BPD [219,222,223,224,225], whether they are administered intravenously or intratracheally. The intraperitoneal administration of EVs was recently considered in an animal model, again with encouraging results [195]. Since EVs are not living cells, they should carry little risk of tumorigenesis or ectopic colonization. In addition, they do not react differentially and unpredictably (like MSCs) to different environments, so they should be more consistent as a therapeutic tool. EVs also have the advantage over MSCs of being cheaper and easier to manage (i.e., to isolate, store and administer) [224]. Our knowledge of their biology and biodistribution, and of the interaction between EVs and target cells is still insufficient, however. Clinical experience is limited to a few trials on the use of EVs derived from dendritic cells in adoptive immune therapies for cancer [226,227], and a single case treated with MSC-derived EVs for steroid-resistant GVHD [228]. As we write, one trial (NCT03857841) is actively recruiting preterm newborns delivered at less than 27 WG at high risk of BPD for the purpose of testing the safety and feasibility of intravenous infusions of bone marrow MSC-derived EVs (UNEX−42). Clearly, before any clinical implementation of these interesting nanoparticles in the treatment of BPD, we need to gain a better understanding of their mode of action, assess their safety, and confirm their efficacy. An even more futuristic approach is the engineering of extracellular vesicles; this appears as a new frontier in order to treat inflammatory diseases with more targeted and specific molecules [229,230].

Research performed on clinicaltrials.gov, updated at 5 February 2020, some of these studies preview related follow up studies.

## 15. Future Needs

An important challenge for the future is to identify BPD biomarkers that help us to predict the need for treatment. Then, we can use these novel therapeutic strategies only in those infants who really need them. Genomics, transcriptomics, proteomics, metabolomics, nutritional factors, and maternal antenatal drugs are all worth considering in BPD research and the international debate on this disorder.

The treatments discussed in this paper are at various stages of clinical implementation. “Topical” steroids are already being used in large-scale clinical studies seeking to definitively confirm the very encouraging preliminary results obtained, and hopefully, rule out any associated increase in mortality [173]. This would seem to be the most obvious next step forward in the management of BPD. IGF−1 therapy is already being applied in clinical settings, while we await the results of a large multicenter study to confirm the early enthusiasm for its use to prevent BPD.

Two of the most revolutionary treatments, MSCs and EVs, are still in the preclinical and experimental stages, which must be completed before any translational use can be attempted [231]. The results obtained so far have been remarkable, but they have the important technical limitation of the small size of the animal model involved.

We can logically expect the use of EVs to be safe (as they are an element contained in MSC-based therapies), but this issue has yet to be fully explored in a phase 1 human study. The most appropriate mode (intravenous or intratracheal) and timing of administration, and the optimal dosage also need to be established. A standardized method for generating and harvesting EVs will also be needed to obtain a product for use in human translational applications [232].

In conclusion, BPD experts and caregivers are looking forward to seeing some really revolutionary advances in the treatment of a disease that is well known, and for which there are numerous evidence-based options to help in its prevention and improve its symptoms, but no definitive treatment to date.

The development of EV-based therapeutics is considered the most promising next-generation approach to preterm infants at risk of BPD. Such treatment may considerably improve the short- and long-term lung development of affected patients, completely changing the natural history of this disease.

## Figures and Tables

**Figure 1 jcm-09-01539-f001:**
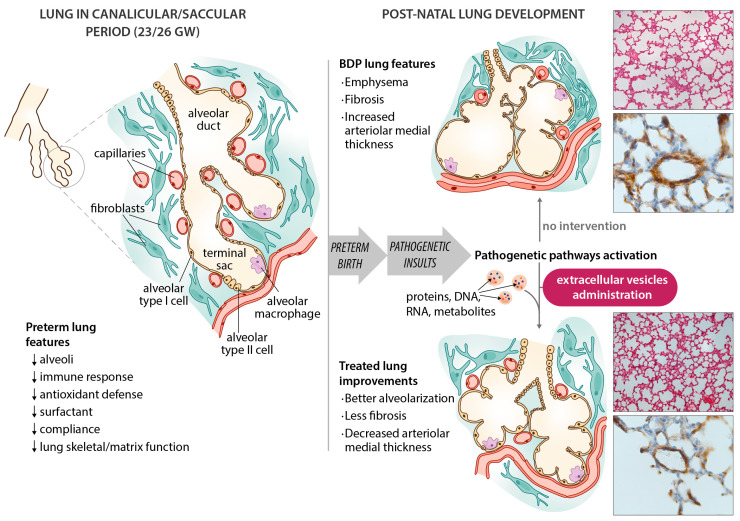
Schematic diagram of bronchopulmonary dysplasia (BPD) pathogenesis, with the main features of the disease in the preterm lung. Microphotographs show improvements in an experimental animal model (60% hyperoxia-induced BPD in rat pups) after administering extracellular vesicles (EVs). Representative lung sections from hyperoxia-induced BPD (top right), and EV-treated animals (bottom right), showing effects of hyperoxia and EVs on alveolarization (upper side of microphotographs; stained with hematoxylin and eosin, scale bars 150 µm) and vessels less than 100 µm in diameter (lower side of the microphotographs; stained with anti-α-smooth muscle actin antibodies, scale bars 23.8 µm). Hyperoxia-induced changes in alveolarization (lower total number of alveoli; larger alveolar volume) and vascularization (increased medial thickness index) are minimized by the intratracheal administration of MSC-EVs.

**Table 1 jcm-09-01539-t001:** BPD prevention strategies.

	BPD Prevention Methods
Antenatal glucocorticoids	Given to women between 23 and 34 WG Less respiratory distress syndrome Shorter periods of mechanical ventilation and oxygen supplementation [120]
Surfactant	Reduces the need for mechanical ventilation and oxygen dependence [121]
LISA technique [122]	Reduces the need for mechanical ventilation [123,124]
Protective ventilation	Low tidal volumes Early weaning from mechanical ventilation Early CPAP and noninvasive ventilation
Targeted O_2_ saturation	Reducesoxidative damage [125]
Early therapy with caffeine	Shorter time on ventilatory support [126] Better lung function [127,128] Modulates angiogenic gene expression early in lung development [129]
Vitamin A	Has a role in lung maturation and repair Reduces the development of BPD at 36 weeks PMA, but has no effect on long-term respiratory morbidity [130,131,132,133,134,135]
Postnatal infection control	Reduces inflammatory mediators and the need for mechanical ventilation
Hemodynamically significant PDA treatment	Reduces pulmonary overflow, and this limits the need for ventilation [136]
Fluid restriction	Prevents pulmonary overflow and consequent lung edema Reduces the incidence of PDA [137]
Azithromycin prophylaxis	In newborns colonized with Urea plasma [138]
Nutrition	Adequate enteral supplement of nutrients [139] Possibly with mother’s own milk [140,141] To ensure a good weight gain [142] L-citrulline in particular seems to correlate with a lower incidence of Pulmonary hypertension [143,144] (an interesting trial [NCT03542812] is ongoing)
Postnatal systemic glucocorticoids	Reduce inflammation, vascular permeability and lung edema Their short- and long-term adverse effects suggest caution in their routine use for preventing BPD [145]

**Table 2 jcm-09-01539-t002:** Ongoing trials with stem cells for BPD.

NCT	Phase	Cells	Route	Dose	Age	Target Enroll-ment	Country
NCT04062136	I	UC-MSCs	Iv	1 × 10^^6^/kg × 2	1–6 months	10	Vietnam
NCT03558334	I	UC-MSCs	Iv	1 × 10^^6^/kg 5 × 10^^6^/kg	nk	12	China
NCT03601416	II	UC-MSCs	Iv	1 × 10^^6^/kg 5 × 10^^6^/kg	up to 1 year	57	China
NCT03645525	I–II	UC-MSCs	It	2 × 10^^7^/kg	up to 3 weeks	180	China
NCT03378063	I	UCB-MSCs	Nk	Nk	1–3 months	100	China
NCT02443961	I	MSCs (not spec)	Nk	5 × 10^^6^/kg × 3	1 month to 28 weeks	10	Spain
NCT03683953	I	MSCs (not spec)	It	25 × 10^^6^/kg	28 to 37 WG	200	China
NCT03631420	I	UC-MSCs	Nk	3 × 10^^6^/kg 10 × 10^^6^/kg 30 × 10^^6^/kg	36–38 WG	9	Taiwan
NCT03774537	I–II	UC-MSCs	Iv	1 × 10^^6^/kg 5 × 10^^6^/kg	3–14 days	20	China
NCT01207869	I	UC-MSCs	It	3 × 10^^6^/kg	up to 6 months	10	Taiwan
NCT02381366	I–II	UCB-MSCs	Nk	10 × 10^^6^/kg 20 × 10^^6^/kg	up to 14 days	12	Illinois–USA
NCT01297205	I	UCB-MSCs	It	10 × 10^^6^/kg 20 × 10^^6^/kg	up to 14 days	9	Republic of Korea
NCT01828957	II	UCB-MSCs	It	10 × 10^^6^/kg	up to 14 days	69	Republic of Korea
NCT03392467	II	UCB-MSCs	Nk	Nk	up to 13 days	60	Republic of Korea
NCT04255147	I	UC-MSCs	Iv	1 × 10^^6^/kg 3 × 10^^6^/kg 10 × 10^^6^/kg	7–21 days	9	Ontario–Canada

Abbreviations: UC: umbilical cord; UCB: umbilical cord blood; MSCs: mesenchymal stem cells; iv: intravenous; it: intratracheal; nk: not known; WG: weeks of gestation.

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
