# Peer review of "Present and Future of Bronchopulmonary Dysplasia"

_jcm, 2020, doi:10.3390/jcm9051539_

Round 1

Reviewer 1 Report

This excellent review by Bonadies and colleagues summarizes the pathogenesis, challenges in the diagnosis and management, and future directions of bronchopulmonary dysplasia (BPD). Although quite comprehensive and timely, the review can be updated further to provide a balanced view of all the pathogenic factors of BPD.

  • Abstract:

Line 14: The authors should consider modifying this sentence. BPD remains the most common respiratory morbidity of preterm infants. The term “nowadays” may be misleading.

  • IUGR and BPD

The authors need to provide references and update the mechanisms through which IUGR increases BPD risk.

  • Preeclampsia and BPD

The discussion on preeclampsia and BPD is quite simplified. The role of preeclampsia in BPD pathogenesis is debatable – it may depend upon associated morbidities such as IUGR.

  • Whole-exome sequencing

Please include and discuss the recent study on the whole-exome sequencing in preterm infants (PMID: 30342483).

  • Additional risk factors                                                                                                                                       The authors need to discuss other factors, including maternal smoking, placental pathology, mode of delivery, multiple gestation, microbiome, and malnutrition in BPD pathogenesis.

Reviewer 2 Report

Comments to the author

          The authors are reviewing a present understanding of the future prospects for the prevention and treatment of bronchopulmonary dysplasia (BPD), based on knowledge of the disease’s pathogenesis, and on the current management of extremely premature infants. This review is well-written.

          There are some concerns in this manuscript as listed below.

Minor concerns

  1. Background section, Page 2, Line 42: One of reasons why the burden and incidence of BPD has remained unchanged is that most (almost all) of extremely premature infants, who were supposed to die long time ago, survive and suffer from BPD now. The authors should state this with some evidences.
  2. Background section, Page 2, Line 45: The authors mentioned incidence of BPD from 15% to 35%. The authors should also mention which definition of BPD was used to calculate its incidence.
  3. Background section, Page 2, Line 56, Prenatal risk factors section, Page 4, Line 141, Long-term pulmonary outcome section, Page 8, Line 297: “The Authors” should be “The authors”.
  4. Background section, Page 2, Line 56-60: The authors are stating an inverse relationship between mortality risk and gestational age. They should also state a relationship between BPD and mortality or long-term morbidities, because they mention that “BPD is also a long-lasting disease, …..” at the beginning (Lane 54).
  5. Definition and Diagnosis section, Page 2, Line 59: The words “offormer” should be “of former”.
  6. Definition and Diagnosis section, Page 3: The authors should discuss when BPD should be diagnosed (e.g. 36 weeks or 40 weeks) using a reference, Isayama T, et al. JAMA Pediatr. 2017;171(3):271-279.
  7. Pathology section, Page 3, Line 98-106: The authors are explaining the pathological findings of BPD. However, no references are observed and I don’t know if these findings are from human or animal tissues. The authors should add some references and explain it.
  8. Demographic risk factors at birth section, Page 4, Line 157: Please add a reference, Ito M, et al. Pediatr Int. 2017;59(8):898-905 to reference #36.
  9. Cellular modifications, Page 5, Line 198: The words “chronic lung disease” should be changed to “BPD”.
  10. Growth factor alterations section, Page 6, Line 220-229. The authors should state an ongoing clinical trial using IGF-1 by TAKEDA, “A Clinical Efficacy and Safety Study of SHP607 in Preventing Chronic Lung Disease in Extremely Premature Infants”.
  11. Table 1, Page 7: Please add a reference, Araki S, et al. PLoS One. 2018;13(11):e0207730 to reference #107-111.
  12. Long-term pulmonary outcome section, Page 8: The authors should discuss the results from papers, Hirata K, et al. Pediatr Pulmonol. 2017;52(6):779-786. and Hirata K, et al. Arch Dis Child Fetal Neonatal Ed. 2015;100(4):F314-9.
  13. New horizons section, Page 9, Line 347-351: The authors should mention an ongoing international clinical trial (https://clinicaltrials.gov/ct2/show/NCT03253263?term=IGF-1&cond=Bronchopulmonary+Dysplasia&draw=2&rank=2).
  14. Table 2, Page 11: The authors should mention that this table came from the website ClinicalTrials.gov., and add in which country these clinical trials conducted or are conducting.
  15. New horizons section, Page 12, Line 435-446: This paragraph should be deleted, because this review is for BPD, not other complications.
  16. Future needs, Page 12, Line 450: Please add “metabolomics” just after “proteomics”.
  17. Figure, Page 13: Scale bars are not visible in the microphotographs. I don’t think these microphotographs are necessary for this figure. Only schematic diagrams are enough.

          Overall, I think this manuscript seems to be suitable, if these all concerns are addressed.
